# Lack of knowledge of stakeholders in the pork value chain: Considerations for transmission and control of *Taenia solium* and *Toxoplasma gondii* in Burundi

**Salvator Minani** [1,2,3]*, **Jean-Bosco Ntirandekura** [4], **Koen Peeters Grietens** [3], **Anastasie Gasogo** [1], **Sarah Gabriël** [2], **Chiara Trevisan** [3]

**1** Department of Biology, Faculty of Sciences, University of Burundi, Bujumbura, Burundi, **2** Laboratory of Foodborne Parasitic Zoonoses, Department of Translational Physiology, Infectiology and Public Health, Faculty of Veterinary Medicine, Ghent University, Merelbeke, Belgium, **3** Department of Public Health, Institute of Tropical Medicine, Antwerp, Belgium, **4** Department of Animal Health and Productions, Faculty of Agronomy and Bio-Engineering, University of Burundi, Bujumbura, Burundi

☸ These authors contributed equally to this work.
* salvator.minani@ub.edu.bi, sminani@itg.be

## Abstract

*Taenia solium* and *Toxoplasma gondii* are important foodborne zoonotic parasites that cause substantial health and economic impacts worldwide. In Burundi, there was a lack of data on the knowledge, attitudes, and practices of stakeholders in the pork value chain. To fill this gap, this study aimed to assess the knowledge of stakeholders in the pork value chain regarding *T. solium* and *T. gondii* infections and identify health-seeking routes and factors contributing to parasitic transmissions. A mixed methods study was conducted in Bujumbura city, Kayanza, and Ngozi provinces from January 10 to April 27, 2023. Quantitative data was collected using questionnaire-based interviews with 386 participants, while qualitative data was gathered from 63 participants through focus group discussions, informal conversations, and observations. The majority of the participants in the quantitative study had heard about porcine cysticercosis (94.8%) and pork tapeworm (90.9%), although the transmission and symptoms of these diseases were less known (>60%) and inaccurately described. Most participants were not aware of human cysticercosis (96.4%), its association with epilepsy (78%), and *T. gondii* infections (91.2%). There was a low proportion of medical consultations for pork tapeworm (30.1%), epilepsy (36.5%), and toxoplasmosis (7%). The qualitative study supported the findings of the quantitative study, revealing low knowledge among participants and misconceptions about the causes, consequences, and treatment-seeking routes related to *T. solium* and *T. gondii* infections. The short roasting time of pork (<15 minutes) and low perception of the consequences of consuming pork infected with cysts exposed pork consumers to these parasitic infections. Inadequate knowledge about these parasitic infections, along with inadequate practices in

**Data availability statement:** All relevant data are within the paper and its Supporting Information files.

**Funding:** This work was supported by the Directorate General for Development Cooperation (DGD), Belgium, through the individual sandwich PhD scholarship programme (Salvator Minani) of the Institute of Tropical Medicine Antwerp, Belgium.

**Competing interests:** The authors have declared that no competing interests exist.

treatment-seeking and pork preparation and consumption, can contribute to continued transmission and pose significant barriers to control programmes. Training and public health education following the One Health approach are urgently needed to better tackle these parasitic infections in Burundi.

## Introduction

Around the world, foodborne parasitic diseases cause significant morbidity, mortality, and economic losses [1]. *Taenia solium* and *Toxoplasma gondii* are ranked among the major foodborne parasites causing substantial health and economic burdens [2,3]. *Taenia solium* is a zoonotic parasite affecting humans and pigs [4]. Pigs develop porcine cysticercosis by ingesting human faeces, feed, or water contaminated with eggs from human *T. solium* tapeworm carriers [5]. Pork tapeworm eggs excreted in human stool may be transmitted to pigs in the case of inappropriate pig husbandry practices and sanitation [6,7]. Larvae settle in the muscles and organs of pigs and become metacestodes (cysticerci) [5,8]. In humans, *T. solium* leads to taeniosis by ingestion of undercooked or raw pork infected with cysticerci, while ingestion of *T. solium* eggs primarily via contaminated water, food, and inadequate hygiene can result in human cysticercosis [8]. The migration of larvae in the human body to the tissues of the central nervous system causes neurocysticercosis (NCC), of which epilepsy is the main clinical sign [5,8,9]. In sub-Saharan Africa, it has been reported that about 30% of people with epilepsy were linked to NCC [10]. A prevalence of 31.5% for human cysticercosis by antibody enzyme-linked immunosorbent assay, 0% to 1% for human taeniosis by microscopy, and 15.5% for cysticercosis in pigs by tongue palpation was reported in Burundi [11–13].

*Toxoplasma gondii* is a cosmopolitan zoonotic parasite, of which cats and other species of the Felidae family are the definitive host and wild and domestic animals, including humans, are the intermediate host [14,15]. Pigs, like other livestock, get infected through ingestion of water or food contaminated with oocysts or via ingestion of meat infected with cysts [15,16]. Humans can acquire toxoplasmosis by ingesting the oocysts via water or food contaminated with cat faeces, by ingesting raw or undercooked meat containing *T. gondii* cysts from livestock and wild animals; blood transfusion or organ transplantation; transplacental transmission and accidental inoculation of tachyzoites [15,17]. Studies reported that 30–63% of acute infections in pregnant women are associated with the consumption of undercooked meat [18]. Human toxoplasmosis is often asymptomatic, but severe cases can include abortion, stillbirth, and the birth of apparently healthy babies who will develop neurological and ocular disorders later in life [19,20]. Neurological and ocular disorders may include encephalitis, mental retardation, hydrocephalus, intracranial calcification, blindness, and chorioretinitis [20,21]. Ten to 20% of immunocompetent patients with toxoplasmosis may show chorioretinitis, lymphadenitis, and polymyositis [21]. In Burundi, the prevalence of toxoplasmosis was 44.1% in humans by Ab-ELISA and indirect immunofluorescence test and 17.7% in pigs by indirect Ab-ELISA [22,23].

Lack of meat inspection, hygiene and sanitation, and some inappropriate practices associated with animal husbandry, pig slaughter, and culinary behaviour are common risk factors for both *T. solium* infections and toxoplasmosis [6,24,25]. Community health education was suggested as an essential part of any intervention to control *T. solium* infections and toxoplasmosis [6,26]. Although community education and awareness campaigns are crucial for disease control, studies conducted in Mexico and Tanzania showed that improving community knowledge does not automatically change common practices and behaviours [27,28]. People might only change their practices after realising that their behaviour can harm their health and cause serious illness, or when economic benefits are threatened [6,27]. Therefore, the lack of knowledge about these diseases, their transmissions, and their clinical manifestations, combined with cultural and community beliefs, could hinder the control of these diseases. A study evaluating the knowledge, practices, and attitudes of stakeholders in Burundi's pork value chain could provide valuable insights for implementing disease control measures. This study aimed to describe the knowledge of stakeholders in the pork value chain regarding *T. solium* and *T. gondii* infections, to contribute to the identification of their specific treatment-seeking routes, and to explore factors contributing to parasitic transmissions.

## Materials and methods

### Ethics statement

The research protocol was approved by the National Ethics Committee in Burundi (CNE/25/2022) and the Institutional Review Board of the Institute of Tropical Medicine (ITM) in Belgium (IRB/RR/AC/091 Ref 1595/22). Local community leaders and study participants provided written informed consent before starting the study. All participants were approached with courtesy and informed about the purpose and importance of the study, types of questions, and themes before the interview and focus group discussions (FGD). They were explained that participation was voluntary and that they had the right to participate, decline, and interrupt the participation at any time without consequences. Verbal consent for the audio records for the FGD was also requested before starting the discussions. Unique coded identifiers were assigned to respect participants' confidentiality and privacy. Additional information regarding the ethical, cultural, and scientific considerations specific to inclusivity in global research is included in the Supporting Information (S1 Checklist).

### Study area

This research was conducted in Bujumbura city, Kayanza, and Ngozi provinces from January 10 to April 27, 2023 (Fig 1). To enrol study participants, pig slaughterhouses, pork markets, and pig farms located in the study area were visited. Bujumbura city was selected being the economic capital and the largest city housing the national slaughterhouse. A substantial number of pigs from the countryside are transported to the city for slaughter and marketing. In addition, there is a significant demand for pork in the peripheral zones of the city. Kamenge and Kinama (Ntahangwa commune) and Kanyosha (Muha commune) were included as pig slaughter and trading of pork for markets, bars, and restaurants are common in these areas. Ngozi and Kayanza provinces were selected as they are among the four most densely populated provinces with large numbers of pigs raised in extensive systems and endemic for *T. solium* [11,29,30]. Three communes in Ngozi province (Ngozi, Gashikanwa, and Busiga) and three communes in Kayanza province (Kayanza, Gatara, and Muhanga) were included in this study.

### Study design

A mixed-method triangulation design was used to collect concurrently quantitative and qualitative data. These data were collected separately to triangulate, compare, and contrast the results.

### Quantitative strand

**Sampling and data collection.** A sample size for participants was calculated using the formula $n = Z^2 pq/L^2$ [31]. Where n is the required sample size, $Z = 1.96$ is the Z-score corresponding to the standard normal distribution for a 95%

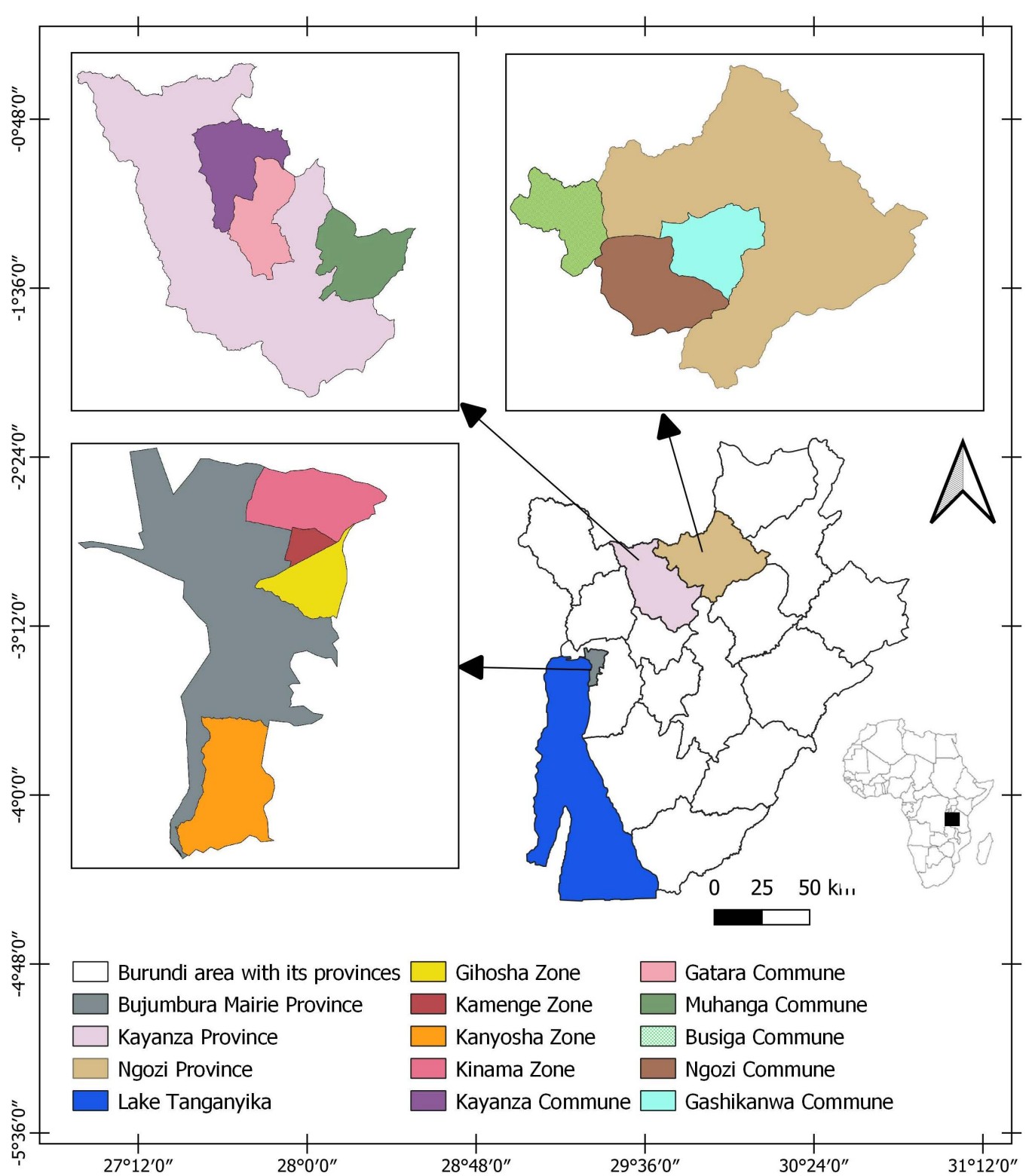

**Fig 1. Study area including Bujumbura city, Kayanza, and Ngozi provinces.**

confidence level, p is the estimated proportion of the study population, q = 1-p (the complement of p), and L is the margin of error, set at 5% (0.05). Since no previous studies have assessed knowledge about *T. solium* and *T. gondii* infections in the study population, we assumed the study population proportion (p) to be 50% (p = 0.5). Thus, 384 study participants were selected from different stakeholders involved in the pork value chain in the study area. A multisampling strategy was adopted to reach out to the stakeholders. Inclusion criteria for the study were established, which required participants to be involved in the pork value chain (e.g., pig farmer, pig trader, butcher, slaughterhouse worker, veterinarian, food safety quality control officer, and pork consumer), be over 18 years old, and willing and able to provide informed consent. A questionnaire was developed and entered into the Epicollect5 application for data collection on a tablet. The survey included both closed and open-ended questions on knowledge regarding *T. solium* and *T. gondii* infections, practices related to treatment-seeking, pork consumption, and culinary methods. The questionnaire was tested before data collection to assess whether it was clear, reliable, valid, understandable, and easy to follow with the Epicollect5 software (https://five.epicollect.net/). Stakeholders taking part in the interviews were selected using random (farmers, butchers), purposive (veterinarians), and convenience sampling (pork consumers). For pig slaughterhouses and pig slaughter slabs, a brief explanation of the study was given to the slaughterhouse manager, who then introduced the researcher to slaughterhouse workers and pig traders. Local community leaders provided a list of pig farmers and butchers, or pork retailers, from which participants were randomly selected. Pork consumers were recruited by convenience in the places where the pork was marketed. Seven veterinarians and two food safety quality control officers working in the study area were purposely recruited. During the interview, each participant was asked to reply to the structured questionnaire. The interviews were made in Kirundi (local language) and responses were then translated into English when entering into Epicollect5. The interview and the translation of responses from Kirundi to English were conducted by the first author. The interview lasted approximately 20 minutes. After each interview, the participant's responses were directly saved on a cloud-based drive.

**Data analysis.** All interview data were validated before analysis by checking for accuracy, reliability, and consistency. Responses were cross-checked for discrepancies, and incomplete or missing data were identified and cleaned. Data were downloaded as a comma-separated values (CSV) file and then exported to R software for analysis [32]. Descriptive statistics were performed to estimate the frequencies and percentages of responses. For all categorical variables, a Chi-square test was performed to estimate associations between them. In addition, a multivariate analysis using a logistic regression model was conducted to identify significant variables contributing to knowledge and practice regarding *T. solium* and *T. gondii* infections and their transmission. The explanatory variables considered in the model included location (urban/countryside), age (≤30/ >30 years), gender (male/female), education (illiterate/literate), occupation (working in the animal sector/working in other sectors), and family size (small/large). Variables were considered significant if $p < 0.05$.

### Qualitative strand

**Sampling and data collection.** Data were collected using focus group discussions (FGD), informal conversations, and observations. Participants in FGDs were selected using a purposive and snowball sampling technique. The same inclusion criteria were applied as for the quantitative data collection. However, participants who took part in the questionnaire interviews were excluded. We ensured that all dimensions of the research were covered and that data saturation, a key indicator for participant recruitment in qualitative research, was reached. Heterogeneity including gender, age range, and education level, was considered when selecting FGD participants, even though some stakeholders declined to participate due to fear of participating or answering the questions. Participants were invited to join the FGD at the agreed place and time. Two facilitators in each location were recruited to help the first author during the discussion for moderation and group management. The number of participants in each FGD varied from 8 to 9 people. Two FGDs were conducted in Bujumbura city, Kayanza, and Ngozi provinces, respectively, resulting in a total of six FGDs. Topic guides were designed before data collection and were updated according to the local context of the study area and the

participants. The FGDs addressed three themes, including (i) knowledge of pork tapeworm and toxoplasmosis infections in pigs and humans, (ii) risk behaviour in culinary practices of pork preparation and consumption, and (iii) knowledge of control methods for pork tapeworm and toxoplasmosis. All FGDs were conducted in Kirundi. They were audio-recorded and lasted approximately 65 minutes. The discussion was closed when data saturation was reached. Conclusions on themes, salient points, and participants' body language were written down by the first author.

Observations and informal conversations were also conducted during the fieldwork. Visits made at different slaughter-houses, pig farms, and pork markets allowed us to observe, interact with people, and understand practices related to pig housing, pig selling, hygiene and sanitation, meat inspection, pig slaughter at home and public places, pork marketing and meat storage and consumption.

**Data analysis.** Audio recordings from the FGDs were transcribed verbatim into Microsoft Word in Kirundi and later translated into English by the first author. Field notes taken during informal conversations and observations were transcribed each evening after the fieldwork, and then translated into English. All transcribed qualitative data were imported, thematized, and coded using NVivo 11 Qualitative Analysis software (QSR International Pty Ltd, Cardigan UK). In addition, reflexivity was applied to triangulate the FGD findings and notes from informal conversations and field observations to ensure the accuracy of the information and identify discrepancies.

## Results

### Study participants

**Quantitative strand.** In the study area, 386 interviews were conducted, including 134 pork consumers, 122 pig farmers, 96 butchers/retailers, 14 slaughterhouse workers, 11 pig traders, 7 veterinarians, and 2 food safety quality control officers. One hundred and ninety-four participants were recruited in Bujumbura city, 94 in Kayanza province, and 98 in Ngozi province (Table 1). The average age of participants was 36 years old, with variations by location: 33 years in Bujumbura city, 41 years in Kayanza province, and 39 years in Ngozi province. Nearly 75% of the participants were male. Approximately 40% of the participants had no formal education, 43.8% had completed primary school, 14.3% had

Table 1. Socio-demographic characteristics of participants.

| Demographic data | Variables | Bujumbura city | Kayanza | Ngozi | Total | % |
|---|---|---|---|---|---|---|
| Gender | Male | 171 | 57 | 61 | 289 | 74.9 |
| | Female | 23 | 37 | 37 | 97 | 25.1 |
| Age | 18-30 years | 90 | 26 | 32 | 148 | 38.3 |
| | 31-50 years | 94 | 47 | 47 | 188 | 48.7 |
| | >50 years | 10 | 21 | 19 | 50 | 13 |
| Education | None | 53 | 52 | 48 | 153 | 39.6 |
| | Primary | 109 | 27 | 33 | 169 | 43.8 |
| | Secondary | 27 | 13 | 15 | 55 | 14.3 |
| | University | 5 | 2 | 2 | 9 | 2.3 |
| Occupation | Farming | 132 | 84 | 81 | 297 | 76.9 |
| | Public sector employee | 6 | 8 | 8 | 22 | 5.7 |
| | Other occupation | 56 | 2 | 9 | 67 | 17.4 |
| Family size | 1-2 people | 52 | 6 | 6 | 64 | 16.6 |
| | 3-5 people | 86 | 42 | 46 | 174 | 45.1 |
| | 6 people and more | 56 | 46 | 46 | 148 | 38.3 |

%: Percentage

finished secondary school, and 2.3% had attained a university level of education. Nearly 77% of the participants worked in the farming sector, 5.7% in the public sector and the rest (17.4%) had other occupations (traders, masons, drivers, carpenters, students, unemployed, and restaurant waiters). Participants' household sizes ranged from 1 to 12 people, with an average of 5 people.

**Qualitative strand.** Six FGDs, including a total of 51 participants, were conducted in Bujumbura city, Kayanza, and Ngozi provinces (Table 2). Thirty-seven participants (72.5%) were male and 14 participants (27.5%) were female. In addition, 12 informal conversations were conducted with 10 male and 2 female participants: 5 in Bujumbura city, 3 in Kayanza province, and 4 in Ngozi province.

## Knowledge about pork tapeworm infections

**Porcine cysticercosis (PCC).** The majority of the participants (94.8%) had heard of PCC (S1 Table). Among them, 79.2% had heard about it in the community, 16.7% at work and 4.1% at school. Over 93% of the participants reported seeing white nodules in infected pigs, such as in the eyes, meat, and tongue. Also, FGD participants knew well about PCC. They had different ways of describing this disease in pigs. They reported that "Ubupera" (cysts) might have come from the term "Amapera" (guava) due to the grains of the guava fruit, which look like cysts. Cysts "Ubupera" were also described as "nodules or whitish buttons found inside the pork resembling a chigoe flea (Imvunja)" (FGD2, FGD6); "finger millets but bigger than finger millets, and when crushed they are full of liquids" (FGD2); "they are inside muscles similar to the ice and when you break them, small larvae will appear from inside" (FGD3); "and after the pig slaughter, you will find it was full in the whole body (cysts) like sorghum grains" (FGD4). The heart and brain were added as locations for the cysts. However, they said the cysts did not invade the outermost layer of the pig's body and showed that infected pigs were asymptomatic.

"*It is difficult to see cysts without looking through its mouth. Cysts are seen on or under the tongue and inside the eyes; they do not occupy the outermost layer of the pig's body. When the pig is infected, it grows fat on the neck and has an excessive appetite, but the thigh is thin, FGD1.*"

In addition, they perceived huge economic losses from the disease when a farmer sold infected pigs or when pigs showed cysts at the slaughterhouse.

"*They [farmer and butcher] fix the price, but the butcher will tell him that after a check of the tongue, he will not pay the full amount if the pig contains cysts. It can happen that instead of three hundred thousand francs (300,000 BIF) for an adult pig, he will give him only one hundred thousand francs (100,000 BIF), FGD3.*"

Approximately 32% of the participants knew that pigs got PCC by ingesting human faeces, feed, or water contaminated with eggs from pork tapeworm carriers. In contrast, 59.0% of the participants did not know the cause of PCC, and 9.3% had misconceptions about it. In the latter case, 47.1% believed that PCC was transmitted by mating with infected pigs,

**Table 2. Overview of participants in the focus group discussions.**

| Study area | Communes | Hills/quarters | Category of stakeholders | Number of participants | Average age |
|---|---|---|---|---|---|
| Bujumbura city | Ntahangwa | Kinama | Butchers and pork consumers | 9 | 30 |
| | Muha | Kanyosha | Butchers and pork consumers | 9 | 33 |
| Kayanza | Kayanza | Nyabihogo | Pig farmers and pork consumers | 8 | 45 |
| | Gatara | Bihororo | Pig farmers and pork consumers | 8 | 45 |
| Ngozi | Ngozi | Kinyana | Pig farmers and pork consumers | 8 | 45 |
| | Busiga | Mihigo | Pig farmers and pork consumers | 9 | 47 |

32.3% by transplacental transmission, 11.7% by consumption of salty feed, and 8.9% by hot climate. The level of knowledge about PCC was significantly different in the three provinces of the study area ($\chi^2 = 21.7$, $p < 0.001$) (S1 Table). Participants from Ngozi and Kayanza provinces had more knowledge than participants from Bujumbura city. The difference between stakeholder groups was significant ($\chi^2 = 39.9$, $p < 0.0001$) regarding knowledge of the cause of PCC (S2 Table). Veterinarians, food safety quality control officers, pig traders, and slaughterhouse workers had more knowledge than other stakeholders. Based on education level, there was no difference in the knowledge of the cause of PCC ($\chi^2 = 4.9$, $p = 0.086$) (S3 Table). In the multivariate analysis, location ($p < 0.002$), age ($p = 0.0009$) and gender ($p < 0.0001$) were significant (S4 Table). Participants from the countryside (Kayanza and Ngozi), those older than 30 years, and male participants had good knowledge of the cause of PCC.

Despite their high knowledge of PCC in the FGDs, participants also displayed inadequate knowledge of its transmission and held misconceptions.

*"We do not know from where pigs get cysts because they can be born with cysts. If we try to analyse the origin of them when they sell their mother, we can find that a sow has cysts. Rather, we think that it depends on the kind of sow and boar and they will have it in that way, FGD6."*

Additional misconceptions about the cause of PCC were reported, including changing pig housing and specific dietary factors like eating banana bark, taro, and brewery grain residues. However, a few participants knew from their experience in pig farming that letting pigs roam freely could be the cause of PCC.

*"Cysts could attack them when owners let pigs wander in the bush. They can wander in the bush and eat the faeces of infected children or adult persons, FGD5."*

**Pork tapeworm.** Pork tapeworm infection was heard of by the majority of the participants (90.9%). Over 87% of these participants had heard about it in the community, 9.1% had learned it at school and 3.7% had heard about it at the healthcare facilities. In the FGDs, participants described the tapeworm infection based on parasite morphology. The tapeworm is known as "Igifwana", a worm that releases segments "Igihuka (proglottids)" and "Igikangaga" like a worm whose segments look like a flat, whitish dry leaf of *Typha* (cattail) used to make mats. Other explanations were provided for the tapeworm as a long white worm, a white rope, and a rounded worm (a twisted worm) like a long rope that separates into several segments when it exits the human intestine through the anus. However, some parts of the explanations seemed correct and others completely incorrect. The tapeworm is flat instead of a rounded worm. On top of that, they were confused about two types of tapeworms (beef tapeworm and pork tapeworm). Some mentioned that the segments could come out of the anus without going to the toilet, which might be specific to the beef tapeworm (present in Burundi) and *Dipylidium* (never reported in Burundi), while for the pork tapeworm, the segments are in the stool.

*"When it[tapeworm] is its time to leave someone's stomach, whether you go to the toilet or not, it will separate into many pieces of segments and then they will drop down one by one and it can take a long time. I used to see one of our family had that tapeworm. He could be sitting and see some parts getting out through his anus, FGD3."*

Most participants (62.1%) did not know the cause of pork tapeworm infection and cited it was due to inadequate hygiene and drinking unsafe water. Nearly 72% did not know the symptoms people infected with tapeworm would experience. The difference in the level of knowledge about pork tapeworm in the three provinces was significant (S1 Table). Participants from Kayanza province had more knowledge than those from Bujumbura city and Ngozi province. Regarding stakeholder groups, the difference in knowledge of the cause ($\chi^2 = 24.1$, $p = 0.0005$) and symptoms of pork tapeworm ($\chi^2 = 26.7$, $p = 0.0001$) was significant (S2 Table). Veterinarians, food safety quality control officers, and pig farmers had

more knowledge than other stakeholders. The difference between education levels and knowledge of the cause ($\chi^2 = 16.6$, p = 0.0002) and symptoms of pork tapeworm ($\chi^2 = 8.9$, p = 0.011) was also significant (S3 Table). Stakeholders with secondary and university education had more knowledge than others with no or only primary education. Based on the multivariate analysis, location (p < 0.0001) and education (p = 0.0003) were significant (S4 Table). Participants from the countryside (Kayanza and Ngozi) and literate participants with secondary or university education had good knowledge of the cause of pork tapeworm.

In the FGDs, they were not well aware of its transmission. Some of them knew it was due to eating undercooked pork, while others provided the same inaccurate answers as in the questionnaire. The tapeworm infection was not perceived as a severe illness beyond the presence of proglottids in the stool, diarrhoea, anorexia, and abdominal and bone pain experienced by infected people.

"*You become unable to have an appetite while eating food. You feel pains in your stomach and bones. If you feel pain whenever you eat some kinds of food you used to eat without any problem, make sure you have a tapeworm. It can grow and begin to break into segments. This can be a significant sign of tapeworm in you, FGD6.*"

In addition, anal itching and pruritus, specific symptoms of beef tapeworm, were listed but mistakenly attributed to amoebiasis for people eating pork infected with cysts. The confusion between amoebiasis and tapeworm could be due to people who had allergies after eating pork or those suffering from human cysticercosis, despite its symptoms taking several months to appear rather than immediately.

"*I think people should roast pork well, if not, the liquids that are inside the cysts can arise amoebiasis for those who have already eaten undercooked pork. So, when you consume it unroasted or undercooked, nodules and itch can quickly appear on someone's body, FGD2.*"

**Human cysticercosis (HCC).**  Most participants (96.4%) had never heard of HCC. Only 14 out of 386 participants (3.6%) had heard about it at school or healthcare facilities. Nearly all of these 14 participants knew how it was transmitted and what the symptoms were (S1 Table). The difference between stakeholder groups and the cause ($\chi^2 = 6.9$, p = 0.076) and symptoms of HCC ($\chi^2 = 1.9$, p = 0.58) was not significant (S2 Table). No difference was found between education levels and the cause ($\chi^2 = 1.5$, p = 0.466) and symptoms of HCC ($\chi^2 = 3.9$, p = 0.139) (S3 Table). In the FGDs, all participants had no idea about the disease. In the multivariate analysis, no significant differences were observed (S4 Table).

**The link between human cysticercosis and epilepsy.**  Most participants (76.4%) had seen people with epilepsy in their neighbourhood (S1 Table). Over 22% of the participants reported that eating pork infected with cysticerci could lead to epilepsy, while 51.5% were unsure about the cause, 12.7% attributed it to evil spirits (witchcraft), 7.8% to hereditary factors, 3.9% to flatulence emitted by people with epilepsy, and 1.8% to trauma during childbirth. The difference between stakeholder groups and the cause of epilepsy was significant ($\chi^2 = 68.3$, p < 0.001) (S2 Table). Veterinarians, food safety quality control officers, pig traders, and slaughterhouse workers had more knowledge than other stakeholders. There was no difference between education levels and the knowledge of the cause of epilepsy ($\chi^2 = 2.5$, p = 0.284) (S3 Table). In the multivariate analysis, only the variable gender (p < 0.0001) was significant (S4 Table). Male participants had good knowledge about the cause of epilepsy.

During the FGDs, epilepsy was perceived as an incurable disease characterised by sudden falls, uncontrolled movements, and loss of consciousness. Several names were attributed to epilepsy depending on the symptoms and causes. According to the symptoms, "Intandara" is a well-known name in the country, which means something that pops up and leads to convulsions and injuries; "Igikange" which means something that surprises someone and suddenly drops; and "Ibisazi" which means madness. Based on the causes, they listed "Ibisigo, Ibishetani, and Imizimu" referring to evil spirits of ancestors or the forest. Apart from the mental problem, the same causes of epilepsy as during the interview were

mentioned. Most participants did not know the link between pork tapeworm and epilepsy. Those who thought epilepsy was due to eating pork infected with cysts listed chronic headaches, seizures, and itchy skin as clinical signs.

"*We heard that the pork tapeworm could lead to epilepsy. For butchers and those who like eating undercooked or bad roasted pork, cysts can reach the brain, causing epilepsy and an endless headache, FGD6.*"

However, they thought patients showed symptoms quickly, while HCC-associated epilepsy takes a long time to manifest.

## Knowledge about toxoplasmosis

Human toxoplasmosis is less known (91.2%) in Burundi. Only 34 out of 386 participants (8.8%) had heard about it (S1 Table). Of these participants, 19 (55.9%) had heard about it in health facilities, 7 (20.6%) had learned about it at school, and 8 (23.5%) had heard about it in the community from women diagnosed with it. Twenty-four participants (70.6%) mentioned abortion, stillbirth, and babies with an abnormality as symptoms associated with the disease. In the FGDs, human toxoplasmosis was not perceived as a severe disease because most participants had no clue about it. Abortion, stillbirth, and babies with abnormalities still occur in the community, although the causes of these remain unknown. The cause of toxoplasmosis was known by 13 participants (38.2%). The difference between stakeholder groups and the cause ($\chi^2 = 11.4$, $p = 0.043$) and symptoms of toxoplasmosis ($\chi^2 = 11.9$, $p = 0.036$) was significant (S2 Table). Veterinarians, food safety quality control officers, and pig farmers had more knowledge than other stakeholders. No difference was found between education levels and the cause ($\chi^2 = 4.6$, $p = 0.099$) and symptoms of toxoplasmosis ($\chi^2 = 1.4$, $p = 0.488$) (S3 Table). No significant differences were found in the multivariate analysis (S4 Table). In the FGDs, a few participants confirmed having heard it, but their knowledge of the cause and symptoms was limited.

"*I heard it from a woman teacher. She told me that having a cat at home could transmit the disease if it breathes on you, FGD6.*"

## Practices and attitudes about treatment-seeking

**Quantitative data.** For pork tapeworm infection, three possible treatment-seeking routes were identified: (i) visiting health facilities for consultation and treatment, (ii) using traditional medicine (herbs) at home, and (iii) visiting the pharmacy to purchase deworming drugs. Of all participants, 30.1% said that people with tapeworm infections went to health facilities, 58.5% did not know if patients went to medical services and 11.4% did not go to any health facility (S5 Table). In the latter case, 4.5% went to the pharmacy to buy deworming tablets and 95.5% stayed at home and used traditional medicine (herbs). They listed several traditional medicines such as *Vernonia amygdalina* (umubirizi or umupfumya), *Senna occidentalis* (umuyoka or umuyokayoka), *Rumex usambarensis* (umufumbegeti), *Plectranthus barbatus* Andrews (igicuncu), *Eriosema psoraleoides* (umupfunyantoke), *Conyza sumatrensis* (umucutsa) and leaves of *Nicotiana tabacum* (itabi) mixed with banana beer. The difference between stakeholder groups and medical consultation was significant ($\chi^2 = 27.5$, $p = 0.006$) (S6 Table). Veterinarians, pig traders, pig farmers, and slaughterhouse workers consulted medical facilities more than other stakeholders. However, no difference was observed among stakeholders not visiting medical facilities ($\chi^2 = 15.5$, $p = 0.050$) (S6 Table). There was no difference between education level for medical ($\chi^2 = 1.7$, $p = 0.79$) and no medical consultation ($\chi^2 = 0.4$, $p = 0.82$) (S7 Table). In the multivariate analysis, location ($p = 0.0006$) and age ($p = 0.041$) were significant (S4 Table). Participants from the countryside and those older than 30 years demonstrated good practices regarding medical consultation for pork tapeworm infection.

For epilepsy, the participants revealed that people with epilepsy would either visit health facilities, stay at home, or consult traditional healers or priests and pastors. Of all participants, 36.5% reported that people with epilepsy would visit a health facility for medical consultation and treatment (S5 Table). Nevertheless, 31.9% declared that they did not consult health facilities and 31.6% did not know if they went to hospitals or stayed at home. When asked about consulting the traditional healer, 7.3% of the participants said they had done so, while others either did not (9.8%) or did not know (82.9%). Depending on the beliefs about the cause of epilepsy, they suggested that for treatment, home remedies (traditional medicine) and prayers could be combined with medication from the hospital. The difference between stakeholder groups for both medical ($\chi^2 = 25.5$, p = 0.012) and traditional healer consultation ($\chi^2 = 27.9$, p = 0.005) was significant (S6 Table). Veterinarians, pig farmers, and slaughterhouse workers consulted medical facilities more than other stakeholders. However, pork consumers, pig traders, and pig farmers consulted traditional healers more than other stakeholders. The difference was significant between education level and medical consultation ($\chi^2 = 9.6$, p = 0.048) (S7 Table). Stakeholders with secondary and university levels visited medical facilities more than others with no or only primary education. In the multivariate analysis, location (p = 0.004) and age (p = 0.041) were significant (S4 Table). Participants from the countryside and those older than 30 years had good practices for medical consultation for epilepsy. However, no difference was observed between education level and traditional healer consultation ($\chi^2 = 1.8$, p = 0.78) (S7 Table).

Regarding toxoplasmosis, the majority of the participants (93.0%) did not know the routes taken by patients. However, 7.0% of the participants reported that patients had to go to health facilities for treatment and check-ups (S5 Table). They also suggested that patients had difficulty knowing about the disease because there were no clinical signs associated with it. Pregnant women were only informed of this when they were tested during the prenatal check-up. The difference between stakeholder groups and medical consultation was significant ($\chi^2 = 67.8$, p < 0.001) (S6 Table). Veterinarians and pig farmers consulted medical facilities more than other stakeholders. The difference was also significant between education level and medical consultation ($\chi^2 = 53.2$, p < 0.001) (S7 Table). Stakeholders with secondary and university levels visited medical facilities more than others with no or only primary education. Based on the multivariate analysis, location (p = 0.036) and education (p < 0.0001) were significant (S4 Table). Participants from the countryside and literate participants with secondary or university education had good practices for medical consultation for toxoplasmosis.

**Qualitative data.** They reported the same routes as the part above, but they mentioned determining factors in the decision-making for each route (Fig 2). Socio-demographic factors (stigma, beliefs about the cause of the disease, religion, entourage) and healthcare factors (costs of consultation and purchase of drugs, availability of drugs or traditional medicine, accessibility to health structures, to the pharmacy and the traditional healer) contributed to the choice for treatment-seeking.

• **Health facility and pharmacy route**

The choice of the decision could be motivated by the awareness of health and community authorities and testimonials from former patients or their families.

*"When you go to the hospital, they diagnose you. They can know what kind of disease you have through stool and urine examinations. Then, after laboratory results, they provide medicines, FGD2."*

However, the disease perception in the community (epilepsy), poverty, remoteness from the hospital, and ineffective drugs (epilepsy and pork tapeworm) prevent patients from going to health facilities.

*"They stay at home because they lack financial means. Years ago, I used to see those who had epilepsy not able to go to the hospital. These aged persons were against sending these patients to the hospital, FGD2."*

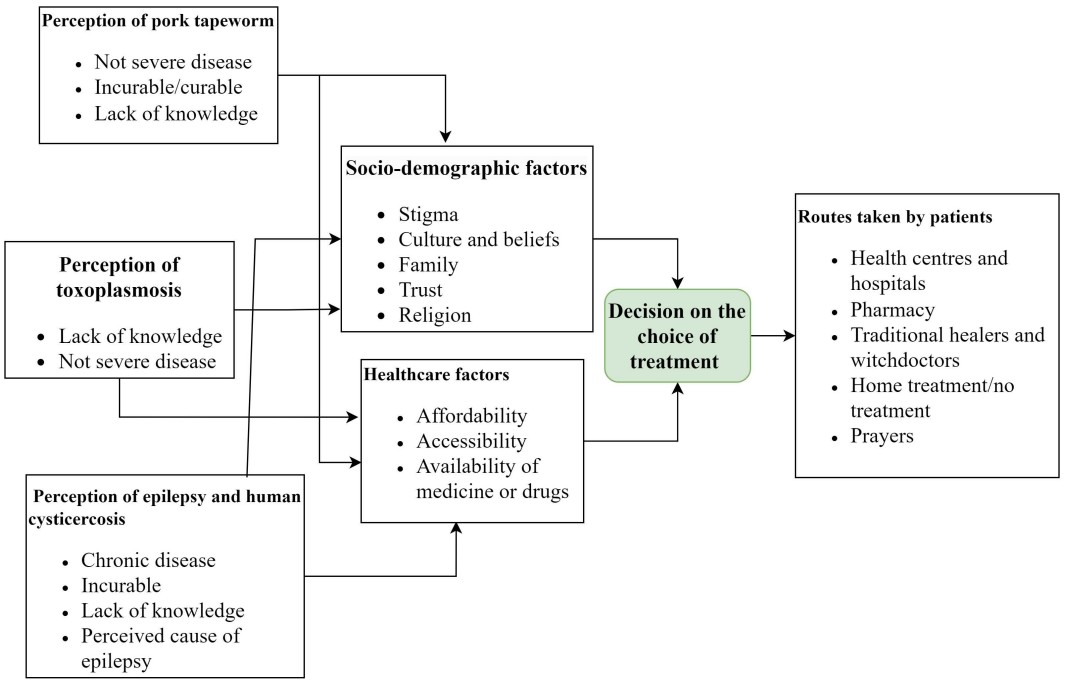

**Fig 2. Flowchart indicating treatment-seeking routes for pork tapeworm, epilepsy, human cysticercosis, and toxoplasmosis and their associated factors in Burundi (Model adapted from the PASS model [33]).**

• **Traditional healers route**

People who believed epilepsy and pork tapeworm infections could not be treated with biomedical drugs often resorted to traditional healers. In Burundi, traditional healers are people who sell different types of herbs at the market or at home to treat many different diseases. In addition, witchdoctors deal with evil spirits and work from home. Trust, the availability of these traditional medicines everywhere in the market, and their cheap price encourage people to use them. However, witchdoctors might recommend frightening practices to people with epilepsy for their treatment, such as holding a dead person in the grave or regularly drinking lamb's urine.

> "*Sometimes they can go to look for witchdoctors after realising that pharmaceutical medicines have been useless. Sometimes, they advise you to hold a dead person and enter first in the grave so, you can be cured of epilepsy, FGD1.*"

• **Stay at home, home treatment and prayers route**

It turned out that sometimes after realising that many treatments requiring money are futile, they seek other ways. Some reported believing in God, praying in church, staying at home, and getting home treatment with forest or field herbs.

> "*There are some people with epilepsy who use other means (prayers and traditional medicine) to mitigate it because we hear people saying that it disappeared completely, FGD4.*"

In addition, they might stay at home due to a lack of awareness about the disease and believe that the disease could not be treated in health facilities, the ineffectiveness of biomedical treatment or to escape stigma from neighbourhood members.

## Practices and attitudes for pork consumption

All participants were asked if they consumed pork. Most participants (93.5%) consumed pork. Of these participants, 32.4% ate pork daily, 37.4% once per week, 11.1% once per month, and 19.1% occasionally per year (once, twice, 3 times, 4 times, and 6 times). Most participants (74.5%) preferred roasted pork and others preferred boiled and fried pork (8.9%), boiled pork (12.2%), and fried pork (4.4%). The difference in pork consumption between the three provinces was significant (S8 Table). The difference between stakeholder groups and pork consumption was also significant ($\chi^2 = 31.7$, $p < 0.001$) (S9 Table). Butchers, slaughterhouse workers, pork consumers, and veterinarians preferred more roasted pork than other stakeholders. No difference based on education level was observed ($\chi^2 = 3.9$, $p = 0.149$) (S10 Table). In the multivariate analysis, a significant difference was observed only for gender ($p < 0.0001$), with females demonstrating good practices of eating boiled or fried pork (S4 Table). The amount of pork consumed ranged from 100 to 500g, with an average of 171g per person, and depended on the financial means of the consumers, especially in the city. Most participants (70.6%) consumed 100-200g per meal (one or two skewers). Roasted pork at bars commonly found in hill centres and quarters of the city was preferred due to cooking time and charcoal saving reasons aligning with consumers' budget. Even though the price of fresh pork per kilo was 11,000BIF (4 USD), one piece of roasted pork weighing 125g was sold at 1,500BIF (0.5 USD). It was also the primary choice for consumers because it was often paired with local drinks. These local drinks are cheap and accessible to people with low resources. However, this method of pork preparation was perceived as unsafe compared to boiling and frying due to its shorter roasting time.

> "*They roast meat well when it is the day. However, in the evening many people will eat it badly roasted. The reason is that many persons need service, and twenty skewers can cost the butcher only ten minutes of roasting, FGD3.*"

They indicated that pork for restaurant and home consumption came from butchers in bars and markets, slaughterhouses, home slaughter by farmers, and clandestine slaughter like stolen pigs during the night and pigs infected with cysts. In the case of pigs infected with cysts, they slaughter and smuggle them to escape the condemnation of carcasses when discovered by a veterinarian during meat inspection. Looking at the consequences of eating pork infected with cysts, the majority of the participants (90.6%) said that it could be inappropriate behaviour that could affect their health (S8 Table). Nevertheless, more than half of these participants (56.0%) mentioned that it could lead to illnesses, but they did not know the specific illnesses that resulted from it. The risk of being infected with taeniosis was answered by 31.7% of the participants, while 10.0% answered for epilepsy and 2.3% did not perceive the risk. The difference in the three provinces was significant ($\chi^2 = 43.4$, $p < 0.0001$) (S8 Table). Based on stakeholder groups, veterinarians, food safety quality control officers, and pig farmers were more aware than other stakeholders that infected pork could cause taeniosis ($\chi^2 = 93.6$, $p < 0.0001$) (S9 Table). This was also noticed for stakeholders with secondary and university levels ($\chi^2 = 25.5$, $p = 0.003$) (S10 Table). In the multivariate analysis, location ($p < 0.0001$), gender ($p = 0.002$), and education ($p = 0.004$) were significant (S4 Table). Participants from the countryside, male participants, and literate participants with secondary or university education had good knowledge of the consequences of eating pork infected with cysts.

However, qualitative data showed that pigs infected with cysts were still consumed, especially those from clandestine slaughter, those slaughtered in the absence of a veterinarian at the slaughterhouse, and even those inspected pigs with light and moderate infections at the slaughterhouse.

> "*When we hear the butcher has found his pig full of cysts, we applaud and say that we are lucky. He will cut according to the money we have, without considering the price of the meat. We do not consider the negative effects of that infected meat, FGD6.*"

Butchers could exacerbate the risk to consumers by selling or roasting infected carcasses overnight because customers would not recognise healthy or unhealthy pork. In addition, the person's body condition (drunkenness), many clients

at bars, clients in a hurry due to hunger, and the limited knowledge of the risk of eating undercooked pork could influence the attitude towards consuming undercooked pork. Some participants suggested that consuming pork infected with cysts might be seen as a preventive measure against amoebiasis.

"…. *Old people will tell you that if you want to eradicate amoeba, it is better to eat pork infected with cysts. So, you will contract thetapeworm, which will consume amoeba, and then it can be treated with no problem, FGD3.*"

Some participants were aware of boiling and frying infected pork to kill all cysts, and so the pork was ready to eat.

"*Depending on the cysts it has, they can cook the meat because boiling water crushes the cysts. If possible, they can filter and pour down the boiled water containing the cysts, then eat the remaining meat, FGD2.*"

## Discussion

This study is the first in Burundi to report the knowledge of stakeholders involved in the pork value chain about *T. solium* and *T. gondii* infections, specific routes of treatment-seeking, and factors contributing to infection transmission.

### Knowledge

Inadequate knowledge about the transmission of *T. solium* and *T. gondii* infections was observed. Most participants had heard about PCC and had seen cysts in pigs, which is in line with findings reported in African and American countries [34,35]. This high knowledge of the participants could be due to cysts seen when performing tongue palpation at pig farms, at pig slaughter, during meat inspection, and when selling meat. However, inadequate knowledge of its transmission in pigs was noted, which is comparable to the results found in Tanzania [36,37]. In addition, very limited knowledge was found in Burkina Faso (0.6%) and South Africa (16%) [38,39]. Similar misconceptions about the cause of PCC, such as mating with infected pigs, transmission at birth, and eating brewery grain residues, were reported in some African countries [38,40]. In agreement with the findings of this study, PCC was perceived in Uganda as a disease that is not clinically manifest but rather identified on examination of the tongue and musculature after slaughter [41]. This study showed that 90.6% of the participants had heard about *T. solium* taeniosis, figures higher than those found in Burkina Faso (53.4%) and Tanzania (48%) [36,42]. Although known by our participants, its prevalence is very low, ranging from 0 to 1% [13]. Nevertheless, inadequate knowledge of its transmission and symptoms was noticed, which is similar to the findings in Tanzania [37]. Moreover, the knowledge of its transmission for this study was higher than in other countries: Uganda (20%), Colombia (6%), and India (21.2%) [43–45]. This study noted the mix-up between tapeworms and roundworms when describing pork tapeworm morphology. The pork tapeworm could not be referred to as the roundworm because it is flat, whitish, and looks like a ribbon with many segments [5]. It is very important that people are able to recognise the morphology of a specific worm, as this could help them describe the symptoms during medical consultation and seek the appropriate treatment. Failure to recognise the symptoms can pose a potential risk of infecting others because the worm is not well treated. Although the Ministry of Public Health organises health awareness campaigns on intestinal worms and their prevention, the complexity of the life cycle of *T. solium* and the fact that most patients remain asymptomatic could also explain the limited knowledge of its transmission for the participants in this study. It was shown that education level could help improve knowledge because the most educated were more knowledgeable about the pork tapeworm than the illiterate participants. Hence continued sensitization about the tapeworm in primary and secondary schools, combined with community education on improved hygiene and sanitation, could be important to prevent *T. solium* infections. Regarding HCC, most participants had never heard about it. This agrees with the findings reported in Zambia, India, and Colombia [35,44,45]. The link between HCC and epilepsy was not well known but the link between pork infected with

cysts and epilepsy was observed. The cause of HCC-associated epilepsy was confusing, where the consumption of pork infected with cysts was attributed to disease transmission instead of ingestion of food and water contaminated with eggs from human *T. solium* tapeworm carriers [8]. Similar misconceptions that epilepsy was caused by evil spirits (witchcraft), hereditary disease, mental problems, and flatulence and saliva from people with epilepsy during seizures were reported in Tanzania, Zambia, and a previous study in Burundi [46–48]. Although misconceptions about the cause of epilepsy were observed, the prevalence of NCC-associated epilepsy in Burundi was estimated at 38.3% by Ag-ELISA [49], which is in line with the figures in Tanzania [50]. These misconceptions about epilepsy could be due to the lack of awareness, lack of diagnostic tools and trained personnel in many health facilities, and negative disease perception. This study showed that the level of education could not change these misconceptions. This is in agreement with the study findings in Tanzania, where some caregivers reported that epilepsy was caused by witchcraft and was contagious through frothing and saliva and by eating fatty foods [46]. Thus, community health education could be important in reducing misunderstandings and misconceptions about epilepsy. This study also revealed limited knowledge of toxoplasmosis among stakeholders involved in the pork value chain. This is consistent with the findings reported in Morocco, Ethiopia, and Mexico for university students, health practitioners, and pregnant women [51–53]. Although university students are expected to have good knowledge, the cause of toxoplasmosis was known to only 36.5% of biology, medicine, and veterinary medicine students in Morocco, which could explain the lack of significant difference between educated and illiterate stakeholders in Burundi [52]. Moreover, 36% of health workers in Tanzania knew the cause and symptoms of toxoplasmosis, while only 4% of pregnant women were aware of it [54]. A slight improvement in knowledge was reported in the United States of America for pregnant women where 40% knew the cause of toxoplasmosis [55]. In Burundi, the lack of information on this disease could be because medical examinations at hospitals were carried out in French, which is not understandable for all. Also, the fact that many health centres in the country, which receive the largest part of the population, could not diagnose toxoplasmosis explains this limited knowledge for our participants. Only people visiting provincial hospitals and major hospitals across the country could be diagnosed with it. Thus, an update of the knowledge of medical and veterinary professionals should be carried out because the health education of the population could not be possible when the trainers in health education have limited and unsuitable knowledge.

### Treatment-seeking routes

The disease perception could be the key for patients to the use of either biomedical drugs or traditional medicine. This is in line with the study findings in Burundi, which described that there were diseases that were treated in (i) health facilities, (ii) both at health facilities and by traditional healers, and (iii) by traditional healers only [56]. It was also reported that spiritual and mental problems, diarrheal diseases, poisonings, and infertility were treated by traditional healers, but malaria was rather reserved for biomedicine [56]. A third of the participants indicated that patients with taeniosis visit health facilities, which shows a low level of medical consultation, possibly because taeniosis does not have serious symptoms [4]. Niclosamide and praziquantel are effective anthelmintic drugs against pork tapeworm infection [57]. However, our participants revealed that the pork tapeworm was incurable. This could be because in most health facilities all intestinal worms are grouped together and treated with the same anthelmintics (mebendazole, albendazole), which at the given dose are not effective for tapeworms [58,59]. If they have been given the correct drug and the correct dose, this belief about the incurability of pork tapeworm could change. Moreover, low sensitivity of the microscopic test for intestinal worms in health facilities and limited patient knowledge of taeniosis could prevent healthcare providers from prescribing the right medication. Using pharmacy medications without being tested for tapeworms could also explain the incurability of the disease by providing inappropriate anthelmintic treatment for taeniosis. Reliance on the use of herbs from traditional healers or the forest was reported because they are cheap, easy to access, and self-privacy is preserved. However, it was reported in Burundi that 55% of the participants consulted a traditional healer and 75% used herbal medicine for self-treatment [56]. The reason given is that traditional healers are flexible in payment, while debts may be incurred in the case of health facility

consultation, which is similar to our findings [56]. Nevertheless, the government of Burundi has implemented massive and free administration of albendazole/mebendazole and praziquantel every year for children under 15 years of age to better prevent and control soil-transmitted helminthiases and schistosomiasis, which could also reduce the prevalence of tapeworm among the children [60]. In addition, low cost, availability, and low toxicity are some of the benefits that attract people to use herbal medicine in Africa [61]. However, its dosage and effectiveness remain a challenge. It was also reported that up to 80% of people in Africa depended on the use of traditional medicine for their health problems [62]. Even though some herbs listed are helpful for gastrointestinal worms, they are not effective for pork tapeworm, except for *Senna occidentalis,* which can be used for tapeworm (*Hymenolepis*) [63–65]. Besides pumpkin seeds which were widely known, some species of Zingiberaceae, Poaceae, Annonaceae, Rubiaceae, Anacardiaceae, and Lauraceae might be effective in treating pork tapeworm, but their effectiveness is questionable and needs to be further elucidated [66]. A similar misconception about pork tapeworm treatment was reported in Uganda, where butchers and pork consumers believed that drinking strong alcohol (waragi) could kill the tapeworm [67]. Therefore, the use of ineffective drugs and herbal medicine leads pork tapeworm carriers to continue spreading the proglottids and eggs in the environment, resulting in the persistence of infection in pigs and humans. When looking at HCC-associated epilepsy, the level of medical consultations for people with epilepsy was low, which is in line with the WHO report in 2019 that more than 75% of people with epilepsy in developing countries do not have access to antiepileptic drugs [68]. However, this low number of medical consultations is due to socio-demographic and healthcare factors that push patients to decide on different treatment routes [33]. The perception of the incurability of epilepsy and misconceptions that epilepsy is caused by evil spirits and witchcraft lead patients to resort first to traditional healers for treatment rather than hospitals [46,47]. While hospitals can serve as a secondary option, the inconsistent availability of drugs, long distances to healthcare facilities, affordability issues related to drug and diagnosis costs, and the shortage of neurology specialists further exacerbate the situation [47]. In Burundi, direct and indirect costs in the hospital setting were estimated at 72 USD in 2020 per case of NCC-associated epilepsy, which is not affordable for everyone [69]. Moreover, stigma, driven by misconceptions about epilepsy being contagious through saliva, urine, and flatulence, as well as community discrimination about work, education, and marriage, could cause people with epilepsy in sub-Saharan Africa to isolate themselves due to fear [70]. In Burundi, 90% of the population is Christian and some churches preach that the use of traditional medicine would be a sin, which explains why the reliance of some participants on prayers and belief in God as the source of deliverance for epilepsy [56]. Nevertheless, religious practices could not prevent followers from resorting to traditional healers and herbal medicine when sick, which explains the trust in traditional medicine without considering some of its side effects [56]. Therefore, public awareness needs to be enhanced to reduce stigma towards people with epilepsy. Moreover, the capacity of health facilities needs to be increased to enable improved service delivery. Although toxoplasmosis is less known, those familiar with it reported that visiting health facilities was crucial for prenatal check-ups and the prevention of transplacental transmission. However, the use of spiramycin, an antiparasitic drug widely used for pregnant women infected with toxoplasmosis, is less effective but could prevent vertical transmission by more than 50% [71]. While abortion is one of the clinical signs of toxoplasmosis, women who sometimes abort may consult traditional healers (witch-doctors), thinking it is due to witchcraft from their enemies in the neighbourhood to show them whether the cause is a supernatural force or not [56]. Thus, public sensitization combined with the extension of diagnostic tools in all health centres and the availability of drugs could help the community to know and prevent toxoplasmosis.

## Pork consumption

Although pork was consumed by participants, a low awareness of the consequences of eating infected pork and inadequate pork preparation and consumption practices was noted. Similar findings were reported in Tanzania and Uganda, where most fried/roasted pork in local bars was undercooked, which could expose consumers to the pork tapeworm [40,41,72]. In agreement with our findings, too many pork orders in bars, lack of pork roasting skills, insufficient knowledge of the consequences of infected pork, and people's preference for how pork is roasted drive people to eat

undercooked pork in Uganda [41]. Furthermore, it was pointed out that drunkenness from local drinks at pork outlets in Uganda and Zambia further contributes to the consumption of undercooked pork, which is similar to the findings in this study [48,67]. However, there are basic guidelines for dealing with infected pork, even though it is challenging for butchers and pork consumers to implement them because infected pork is still roasted or fried in pork outlets and at home. Cooking was considered as an effective method to control *T. solium* cysts and *T. gondii* cysts in pork. Traditionally boiled pork has been shown to kill *T. solium* cysts at 80°C for over 10 minutes [73]. In addition, temperatures from 60°C to 85°C could kill *T. solium* cysts for pork roasted traditionally for 17 hours and 90°C to 100°C for pork roasted commercially for 3 hours [74]. This cannot be implemented in Burundi because most pork is roasted in less than 15 minutes. While *T. gondii* cysts are difficult to detect with the naked eye, the pork will be safe when it is heated at 67°C for 3 minutes [75]. Although pork fried in oil is supposed to be safe, frying time is very important for thick pork to destroy any pathogens. The fast deep-fried pork, mainly practised in most restaurants and homes in Tanzania, was considered unsafe because of the hurry of pork consumers due to hunger or preference for how pork is fried [73,76]. Thus, health promotion could be substantial for butchers and pork consumers to change behavioural practices regarding the preparation and consumption of pork, although this may take time, to reduce the risk of pork-borne parasitic infections in Burundi. Also, raising public awareness about properly cooking pork could be the best solution to effectively destroy *T. solium* and *T. gondii* cysts before reaching consumers.

## Conclusions

This study revealed inadequate knowledge of stakeholders involved in the pork value chain in Burundi regarding the transmission and symptoms of *T. solium* and *T. gondii* infections, as well as inadequate practices in treatment-seeking and pork preparation and consumption. This limited knowledge, combined with inadequate practices in pork preparation and consumption and treatment-seeking, could pose serious obstacles to the prevention and control of these foodborne parasitic infections. Therefore, the integrated control interventions following the One Health approach are urgently needed in Burundi to improve the knowledge of pork value chain stakeholders and the public through health education and continuing training to better tackle *T. solium* and *T. gondii* infections.

## Supporting information

**S1 Checklist. PLOS' questionnaire on inclusivity in global research.**
(DOCX)

**S1 Appendix. Survey questionnaire (quantitative study) and topic guide (qualitative study).**
(DOCX)

**S2 Appendix. Raw data for quantitative study.**
(CSV)

**S1 Table. Participants' knowledge of *T. solium* and *T. gondii* infections.**
(DOCX)

**S2 Table. Knowledge of *T. solium* and *T. gondii* infections based on stakeholder groups.**
(DOCX)

**S3 Table. Knowledge of *T. solium* and *T. gondii* infections based on the education level.**
(DOCX)

**S4 Table. Multivariate analysis using logistic regression model.**
(DOCX)

**S5 Table. Treatment-seeking practices and attitudes.**
(DOCX)

**S6 Table. Treatment-seeking routes based on stakeholder groups.**
(DOCX)

**S7 Table. Treatment-seeking routes based on the education level.**
(DOCX)

**S8 Table. Practices for pork consumption.**
(DOCX)

**S9 Table. Pork consumption and preparation based on stakeholder groups.**
(DOCX)

**S10 Table. Pork consumption and preparation based on the education level.**
(DOCX)

## Acknowledgments

The authors greatly thank all participants in this study. We would also like to thank the local administration and communal and provincial officers for facilitating the fieldwork and the completion of this study.

## Author contributions

**Conceptualization:** Salvator Minani, Jean-Bosco Ntirandekura, Anastasie Gasogo, Sarah Gabriël, Chiara Trevisan.

**Data curation:** Salvator Minani.

**Formal analysis:** Salvator Minani.

**Funding acquisition:** Salvator Minani, Sarah Gabriël, Chiara Trevisan.

**Investigation:** Salvator Minani.

**Methodology:** Salvator Minani, Sarah Gabriël, Chiara Trevisan.

**Project administration:** Salvator Minani, Sarah Gabriël, Chiara Trevisan.

**Resources:** Salvator Minani.

**Software:** Salvator Minani.

**Supervision:** Jean-Bosco Ntirandekura, Koen Peeters Grietens, Anastasie Gasogo, Sarah Gabriël, Chiara Trevisan.

**Validation:** Salvator Minani, Koen Peeters Grietens, Sarah Gabriël, Chiara Trevisan.

**Visualization:** Salvator Minani.

**Writing – original draft:** Salvator Minani.

**Writing – review & editing:** Salvator Minani, Jean-Bosco Ntirandekura, Koen Peeters Grietens, Anastasie Gasogo, Sarah Gabriël, Chiara Trevisan.

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
