## [Decision Letter · Decision Letter 0]

Dear Dr. Minani,

Thank you for submitting your manuscript to PLOS ONE. After careful consideration, we feel that it has merit but does not fully meet PLOS ONE’s publication criteria as it currently stands. Therefore, we invite you to submit a revised version of the manuscript that addresses the points raised during the review process.

**ACADEMIC EDITOR: Please insert comments here and delete this placeholder text when finished.**

TITLE

Lack of knowledge of stakeholders in the pork value chain: considerations for transmission and control of Taenia solium and Toxoplasma gondii in Burundi

Review Report

This paper reports the level of knowledge, health-seeking behaviour and risk factors for prevalence of infections with Taenia solium and Toxoplasma gondii in Burundi. The researchers used a mixed-method approach to capture as much information as possible and validate it through triangulation of the information. This article makes significant contribution to the existing body of knowledge in these neglected parasitic zoonoses in Burundi and Sub-Saharan Africa at large.

To improve the reporting of this piece of research, the authors should consider the following comments based on my review of the manuscript:

ABSTRACT

• Despite the use of a combination of quantitative, qualitative, and observational methods of data collection, the abstract reports only qualitative information. I advise the authors to report some key quantitative results in the abstract as well.

• The sample size information for the qualitative study should also be provided in the abstract.

• Results provided in the abstract should indicate their respective source/basis (quantitative versus qualitative study).

INTRODUCTION

The authors should provide information on the prevalence of Taenia solium and Toxoplasma gondii in Burundi. Currently, no mention at all. If they are not prevalent, the current study lacks a foundation.

MATERIALS AND METHODS

Quantitative strand - sampling and data collection

• The sample size formula for the quantitative study provided is not elaborated. Authors should (i) define the letters included in the formula and (ii) the values they have used to arrive at the 386 people.

• The authors said “All interview data were validated before analysis”. They should explain how the validation was done.

• Could running a multivariate analysis add value to the research findings?

Qualitative strand

• Determination of sample sizes for the various components (FGDs, informal conversation, and observations) is not described.

• Qualitative data analysis is not clearly described.

RESULTS

• Authors should double-check the conformance of the manuscript with the PLOSONE format requirement. I find specifically that the vertical and several lines in the tables might not conform to the journal’s format. Similarly, some tables are too long.

• As for porcine cysticercosis, please mention the local name for toxoplasmosis.

publication criteria  and not, for example, on novelty or perceived impact.

We look forward to receiving your revised manuscript.

Kind regards,

Khaled Abd EL-Hamid Abd EL-Razik, Ph.D.

Academic Editor

PLOS ONE

2. Please include the following request in the decision letter, and ping me with follow up. “Please include a complete copy of PLOS’ questionnaire on inclusivity in global research in your revised manuscript. Our policy for research in this area aims to improve transparency in the reporting of research performed outside of researchers’ own country or community. The policy applies to researchers who have travelled to a different country to conduct research, research with Indigenous populations or their lands, and research on cultural artefacts. The questionnaire can also be requested at the journal’s discretion for any other submissions, even if these conditions are not met.  Please find more information on the policy and a link to download a blank copy of the questionnaire here: https://journals.plos.org/plosone/s/best-practices-in-research-reporting. Please upload a completed version of your questionnaire as Supporting Information when you resubmit your manuscript.

3. Thank you for stating the following financial disclosure:  [This work was supported by the Directorate General for Development Cooperation (DGD), Belgium, through the individual sandwich PhD scholarship programme (Salvator Minani) of the Institute of Tropical Medicine Antwerp, Belgium.].  Please state what role the funders took in the study.  If the funders had no role, please state: "The funders had no role in study design, data collection and analysis, decision to publish, or preparation of the manuscript." If this statement is not correct you must amend it as needed. Please include this amended Role of Funder statement in your cover letter; we will change the online submission form on your behalf.

5. Please include a caption for figures 1 and 2.

Additional Editor Comments:

TITLE

Lack of knowledge of stakeholders in the pork value chain: considerations for transmission and control of Taenia solium and Toxoplasma gondii in Burundi

Review Report

This paper reports the level of knowledge, health-seeking behaviour and risk factors for prevalence of infections with Taenia solium and Toxoplasma gondii in Burundi. The researchers used a mixed-method approach to capture as much information as possible and validate it through triangulation of the information. This article makes significant contribution to the existing body of knowledge in these neglected parasitic zoonoses in Burundi and Sub-Saharan Africa at large.

To improve the reporting of this piece of research, the authors should consider the following comments based on my review of the manuscript:

ABSTRACT

• Despite the use of a combination of quantitative, qualitative, and observational methods of data collection, the abstract reports only qualitative information. I advise the authors to report some key quantitative results in the abstract as well.

• The sample size information for the qualitative study should also be provided in the abstract.

• Results provided in the abstract should indicate their respective source/basis (quantitative versus qualitative study).

INTRODUCTION

The authors should provide information on the prevalence of Taenia solium and Toxoplasma gondii in Burundi. Currently, no mention at all. If they are not prevalent, the current study lacks a foundation.

MATERIALS AND METHODS

Quantitative strand - sampling and data collection

• The sample size formula for the quantitative study provided is not elaborated. Authors should (i) define the letters included in the formula and (ii) the values they have used to arrive at the 386 people.

• The authors said “All interview data were validated before analysis”. They should explain how the validation was done.

• Could running a multivariate analysis add value to the research findings?

Qualitative strand

• Determination of sample sizes for the various components (FGDs, informal conversation, and observations) is not described.

• Qualitative data analysis is not clearly described.

RESULTS

• Authors should double-check the conformance of the manuscript with the PLOSONE format requirement. I find specifically that the vertical and several lines in the tables might not conform to the journal’s format. Similarly, some tables are too long.

• As for porcine cysticercosis, please mention the local name for toxoplasmosis.

Reviewers' comments:

Reviewer's Responses to Questions

**Comments to the Author**

1. Is the manuscript technically sound, and do the data support the conclusions?

Reviewer #1: Yes

Reviewer #2: Yes

2. Has the statistical analysis been performed appropriately and rigorously?

Reviewer #1: Yes

Reviewer #2: No

3. Have the authors made all data underlying the findings in their manuscript fully available?

Reviewer #1: Yes

Reviewer #2: Yes

4. Is the manuscript presented in an intelligible fashion and written in standard English?

Reviewer #1: Yes

Reviewer #2: Yes

Reviewer #1: The paper is dealing with two highly threatening zoonotic diseases. Authors tried to figure out and stand on the basic knowledge at stakeholders about both diseases. I believe that they obtained valuable information about the education and learning attitude of them about the diseases. Experiments, statistics, and other analyses are performed to a high technical standard and are described in sufficient detail. Conclusions are presented in an appropriate fashion and are supported by the data. The article is presented in an intelligible fashion and is written in standard English. The research meets all applicable standards for the ethics of experimentation and research integrity. The article adheres to appropriate reporting guidelines and community standards for data availability.

So, I agreed with publishing of the manuscript.

Reviewer #2: PONE-D-24-40420

TITLE

Lack of knowledge of stakeholders in the pork value chain: considerations for transmission and control of Taenia solium and Toxoplasma gondii in Burundi

Review Report

This paper reports the level of knowledge, health-seeking behaviour and risk factors for prevalence of infections with Taenia solium and Toxoplasma gondii in Burundi. The researchers used a mixed-method approach to capture as much information as possible and validate it through triangulation of the information. This article makes significant contribution to the existing body of knowledge in these neglected parasitic zoonoses in Burundi and Sub-Saharan Africa at large.

To improve the reporting of this piece of research, the authors should consider the following comments based on my review of the manuscript:

ABSTRACT

• Despite the use of a combination of quantitative, qualitative, and observational methods of data collection, the abstract reports only qualitative information. I advise the authors to report some key quantitative results in the abstract as well.

• The sample size information for the qualitative study should also be provided in the abstract.

• Results provided in the abstract should indicate their respective source/basis (quantitative versus qualitative study).

INTRODUCTION

The authors should provide information on the prevalence of Taenia solium and Toxoplasma gondii in Burundi. Currently, no mention at all. If they are not prevalent, the current study lacks a foundation.

MATERIALS AND METHODS

Quantitative strand - sampling and data collection

• The sample size formula for the quantitative study provided is not elaborated. Authors should (i) define the letters included in the formula and (ii) the values they have used to arrive at the 386 people.

• The authors said “All interview data were validated before analysis”. They should explain how the validation was done.

• Could running a multivariate analysis add value to the research findings?

Qualitative strand

• Determination of sample sizes for the various components (FGDs, informal conversation, and observations) is not described.

• Qualitative data analysis is not clearly described.

RESULTS

• Authors should double-check the conformance of the manuscript with the PLOSONE format requirement. I find specifically that the vertical and several lines in the tables might not conform to the journal’s format. Similarly, some tables are too long.

• As for porcine cysticercosis, please mention the local name for toxoplasmosis.

**Do you want your identity to be public for this peer review?** For information about this choice, including consent withdrawal, please see our Privacy Policy

Reviewer #1: No

Reviewer #2: No

---

## [Author Response · Author response to Decision Letter 1]

21 Feb 2025

All Reviewer and editor comments and suggestions were addressed (see response to reviewers document). In addition, editor comments related to formatting guidelines, figures (uploaded to PACE), tables, PLOS' questionnaire on inclusivity in global research, data availability and role of the funder in the manuscript were taken into account in the revised manuscript (please see them in the manuscript without changes and with track changes in red color).

---

## [Decision Letter · Decision Letter 1]

Lack of knowledge of stakeholders in the pork value chain: considerations for transmission and control of Taenia solium and Toxoplasma gondii in Burundi

PONE-D-24-40420R1

Dear Dr. Minani,

We’re pleased to inform you that your manuscript has been judged scientifically suitable for publication and will be formally accepted for publication once it meets all outstanding technical requirements.

Kind regards,

Adetayo Olorunlana, Ph.D.

Academic Editor

PLOS ONE

Additional Editor Comments (optional):

Reviewers' comments:

Reviewer's Responses to Questions

**Comments to the Author**

Reviewer #1: All comments have been addressed

Reviewer #2: All comments have been addressed

2. Is the manuscript technically sound, and do the data support the conclusions?

Reviewer #1: Yes

Reviewer #2: Yes

3. Has the statistical analysis been performed appropriately and rigorously?

Reviewer #1: N/A

Reviewer #2: Yes

4. Have the authors made all data underlying the findings in their manuscript fully available?

Reviewer #1: Yes

Reviewer #2: Yes

5. Is the manuscript presented in an intelligible fashion and written in standard English?

Reviewer #1: Yes

Reviewer #2: Yes

Reviewer #1: Authors addressed all comments raised by reviewers. I believe the manuscript is ready for full publication. The manuscript presented in an intelligible fashion. Data included in the manuscript supported the conclusions.

Reviewer #2: I have no further queries to the manuscript as all my previous queries have been properly addressed. It is now well done.

**Do you want your identity to be public for this peer review?** For information about this choice, including consent withdrawal, please see our Privacy Policy

Reviewer #1: No

Reviewer #2: No

---

## [Editor Report · Acceptance letter]

PONE-D-24-40420R1

PLOS ONE

Dear Dr. Minani,

I'm pleased to inform you that your manuscript has been deemed suitable for publication in PLOS ONE. Congratulations! Your manuscript is now being handed over to our production team.

Kind regards,

on behalf of

Associate Professor Adetayo Olorunlana

Academic Editor

PLOS ONE